# Chirality flips of skyrmion bubbles

Yuan Yao [1,4] ✉, Bei Ding[1,4], Jinjing Liang[1,2], Hang Li [1], Xi Shen[1], Richeng Yu[1] & Wenhong Wang [1,2,3] ✉

The investigation of three-dimensional magnetic textures and chirality switching has attracted enormous interest from the perspective of fundamental research. Here, the three-dimensional magnetic structures of skyrmion bubbles in the centrosymmetric magnet MnNiGa were reconstructed with the vector field tomography approach via Lorentz transmission electron microscopy. The magnetic configuration of the bubbles was determined based on the reconstructed magnetic induction (B-field) at their surfaces and centers. We found that the bubbles easily switched their chirality but preserved their polarity to retain their singularity in the matrix of the material. Our results offer valuable insights into the chirality behavior of skyrmion bubbles.

Due to their small size and fast mobility, skyrmions, which are novel topological magnetic structures, have recently become a popular research topic for potential memory device applications[1–4]. In general, skyrmions are stabilized in noncentrosymmetric systems with uniform chirality because of Dzyaloshinskii-Moriya interactions (DMIs)[2,5,6]; however, skyrmions can also be formed in centrosymmetric magnets due to the competition between dipole–dipole interactions (DDIs) and magnetic anisotropy[7–9]. The DMI and DDI skyrmions both have two degrees of freedom, namely, the polarity ($p$) and the vorticity ($w$), that mutually determine the chirality. Considering its stability, the DMI skyrmion is regarded as the best candidate for spintronic applications, although the operation of the DMI skyrmion must reverse the swirl accompanied by its polarity due to intrinsic DMIs[10]. Contrary to DMI-stabilized skyrmions, which exhibit uniform chirality and polarity due to intrinsic DMIs, skyrmion bubbles in centrosymmetric magnets always prioritize individual features and deform randomly because DDIs do not strongly couple the bubbles, resulting in a lack of coordinated behavior. The most interesting physics associated with skyrmion bubbles is that their vorticity of spin texture varies with the internal magnetic structure, resulting in a variety of features[7,8].

Lorentz transmission electron microscopy (LTEM) is a powerful tool for characterizing skyrmions and observing diverse topological configurations with high spatial resolution. In situ techniques, such as cooling, heating, or the addition of an electromagnetic field, can be used to directly observe the transformation and movement of skyrmions via LTEM[11–13]. However, even without the artifacts produced by the specimen orientation or image processing[14,15], the magnetic configurations revealed by the transport of intensity equation (TIE)[6], holography[16,17], or differential phase contrast (DPC) method[18,19] still conceal some features of the skyrmions since the LTEM image is only a projection of the magnetic induction, which includes the stray magnetic field. Figure 1 depicts this shortcoming in an investigation of simulated Bloch skyrmion bubbles with different spin configurations. Two parameters, the polarity ($p$) and vorticity ($w$), can be used to represent the characteristics of the skyrmion bubbles. In a given coordinate system, the spins at the cores of the bubbles can be pointed either up ($p+$) or down ($p-$), while the in-plane magnetization can rotate either clockwise ($w+$) or counterclockwise ($w-$). The polarity of the skyrmion bubbles can be identified by the surface twist, where $p+$ represents a divergent spin arrangement at the top and convergent magnetic twisting at the bottom, while $p-$ represents the opposite scenario. The polarity and vorticity determine the chirality of the skyrmion bubble as L+ ($p+$, $w+$), L− ($p-$, $w-$), R+ ($p+$, $w-$) or R− ($p-$, $w+$), as illustrated in Fig. 1a. The corresponding simulated LTEM images and the retrieved in-plane magnetic configurations are shown in Fig. 1b. Regardless of their chirality, all clockwise skyrmion bubbles ($w+$) display the same magnetic contrast in the LTEM images, and the in-plane magnetic induction components have identical directions; the same characteristics are observed for the counterclockwise skyrmions ($w-$). Thus, typical LTEM images fail to discern the true magnetic configurations of the skyrmion bubbles, and it is difficult to determine whether a bubble varies its chirality based on only the inverse image contrast. Thus, the three-dimensional (3D) magnetic structure should be investigated to thoroughly understand the real features of skyrmions in magnets.

[1]Beijing National Laboratory for Condensed Matter Physics, Institute of Physics, Chinese Academy of Sciences, Beijing 100190, China. [2]University of Chinese Academy of Sciences, Beijing 100049, China. [3]Songshan Lake Materials Laboratory, Dongguan, Guangdong 523808, China. [4]These authors contributed equally: Yuan Yao, Bei Ding. ✉e-mail: yaoyuan@iphy.ac.cn; wenhong.wang@iphy.ac.cn

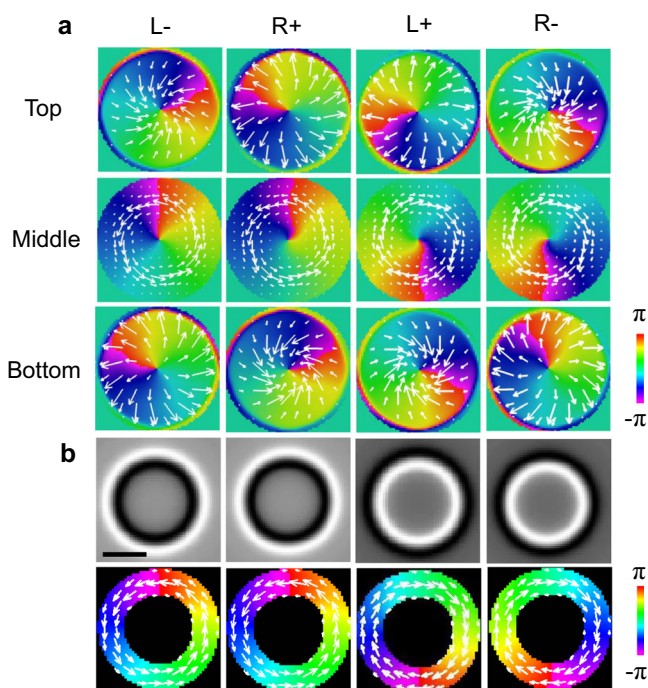

**Fig. 1 | Simulation of skyrmion bubbles. a** Magnetization characteristic at the surfaces and centers of different skyrmion bubbles. **b** The simulated 300 μm overfocused Lorentz TEM images and the recovered in-plane magnetization map from the simulated Lorentz TEM images (the color denotes the direction of the local induction). The contrast below 5% of the maximum magnitude is masked in the magnetization mapping to filter the artifacts induced by the regularization parameter in the TIE. The scale bar is 50 nm. The polarity and vorticity determine the chirality of the skyrmion bubble as L+ ($p+, w+$), L− ($p−, w−$), R+ ($p+, w−$) or R− ($p−, w+$).

Recently, several techniques have been developed to determine the detailed 3D magnetic textures of DMI-induced skyrmions in non-centrosymmetric systems, such as resonant elastic X-ray scattering (REXS)[20,21] and associated scanning X-ray magnetic circular dichroism (XMCD)[22]. In real space, TEM tomography is an appropriate approach for visualizing the 3D morphology of a specimen, yielding important material characterization information[23]. However, in contrast to the typical reconstruction of morphologies or chemical concentrations in real space, where the quantitative contribution of each voxel to the image contrast does not change as the specimen rotates, the 3D recovery of the magnetic structure is more complicated because each vector of the voxel varies its projection on the image at different orientations depending on the dot product between the vector and the projection line[24,25]. Therefore, the traditional algorithm cannot be used to reconstruct the vector field directly. However, the vector component along the tilting axis always provides a constant contribution to the image contrast (the angle between the component along the tilting axis and the projection direction is always a right angle, so its dot product does not change as the specimen tilts); thus, the 3D feature of this component can be retrieved from a series of tilting images with normal scalar reconstruction. Then, the three orthogonal components of the magnetic induction, $B_x$, $B_y$, and $B_z$, can be reconstructed separately by tilting the specimen around the corresponding axes, and the entire vector field can be reconstructed according to these components. Fortunately, because $\nabla \cdot \mathbf{B} = 0$, only two orthogonal components, such as $B_x$ and $B_y$, must be determined to rebuild the whole 3D magnetic vector field with the proper boundary conditions[26–29]. Wolf et al. brilliantly described the entire pipeline in great detail[29]. With this algorithm, some micromagnetic features have been successfully portrayed, including DMI skyrmions[30].

In this work, we acquired multiangle LTEM images around the x- and y-axes to determine the internal 3D magnetic configuration of skyrmion bubbles in the centrosymmetric magnet MnNiGa with this tomography technique. The tomography reconstructions and micromagnetic simulations were combined, revealing the real-space 3D magnetic features of different types of bubbles. Moreover, by analyzing the twisting features on the surfaces of the bubbles, we deduced the chirality of the skyrmion bubbles without obtaining the out-of-plane component. We found that some bubbles easily changed their in-plane components but retained their polarity, resulting in reversed chirality. The reversal of the chirality without reversing the polarity facilitates the application of DDI skyrmions as a feasible memory device.

## Results

A tomography reconstruction of the magnetic configuration was performed on the centrosymmetric magnet $(Mn_{1-x}Ni_x)_{65}Ga_{35}$ (X = 0.45), which was explored in our previous work[31]. MnNiGa was chosen because the field-free magnetic bubbles are stable at room temperature[11,12]. The experimental details are described in the Methods section and Supplementary Information. Figure 2a shows overfocused images and retrieved phase contrast images of the skyrmion bubbles at $\theta_x = 0.4°$ and $\theta_y = 1.2°$. In the LTEM images, the closed spin arrangement represents typical skyrmion bubbles. A few bubbles change their features under the irradiation of the electron beam[32]; thus, a region containing stable bubbles was chosen to perform tilting series imaging. However, some bubbles in the observation area still showed different configurations in the two tilting experiments, even though the bubbles remained unchanged during the respective acquisitions. As shown in Fig. 2b, compared with the bubbles in the x-tilting images, some bubbles in the y-tilting images changed their contrast or shape. The induction orientations deduced from the phase images reveal that the vorticities in some bubbles change, such as bubbles #1 and #9, while some bubbles maintain their initial vorticities, such as bubble #2. It is difficult to confirm which parts of the bubble spins changed with Fig. 2b since changes in the vorticity and polarity can both result in this observed phenomenon. Even if the vorticity and polarity were altered simultaneously, the final state retained the initial features for bubble #2. Thus, the 3D structure of the spin configuration must be explored to determine what occurs in the bubbles. However, the variation in the vorticities and shapes of the bubbles in the y-tilting images rules out the possibility of reconstructing the correct 3D features of those bubbles by directly combining the $B_x$ and $B_y$ components deduced from the individual tilting series.

To extract the characteristics of the bubbles in MnNiGa, we comprehensively analyzed the 3D magnetic induction data. Bubble #2 in Fig. 2, which exhibits little shape change, is optimal for evaluating the feasibility of the vector field reconstruction approach. The rendered orientation of the in-plane magnetization components (Fig. 3a) in three dimensions in the simulated R+ model demonstrates convergent or divergent features on the bubble surface, while the calculated in-plane magnetic inductions are shown in Fig. 3b. The magnetic induction field involving the magnetostatic effect in Fig. 3b replicates the moment style of the R+ bubble. To assess the reliability of the reconstruction, tilt series LTEM images were simulated with the R+ model under the same experimental conditions to rebuild the magnetic induction with the same processing parameters. Figure 3c illuminates the reliable recovery of the R+ bubble; however, some artifacts of the TIE parameters and the missing wedge effect can be observed (weak elongation along the z direction owing to the limited tilting range during data acquisition). Figure 3d outlines the in-plane magnetic induction retrieved for bubble #2, which appears to have a similar contour to the induction shown in Fig. 3c, confirming that bubble #2 is an R+ type bubble. The configurations of the induction

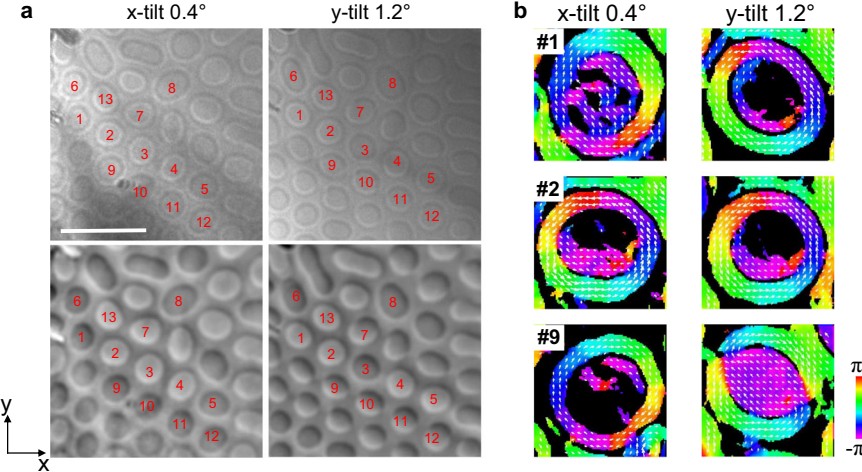

**Fig. 2 | Real-space observation of field-free room temperature skyrmion bubbles after FC manipulation. a** The 300 µm overfocused images (first row) and TIE retrieved phase images (second row) with an x-tilt of 0.4° and a y-tilt of 1.2°, respectively. The scale bar is 500 nm. **b** The configuration of the in-plane magnetic induction of bubbles #1, #2, and #9 was deduced from the phase images in (**a**). The contrast below 5% of the maximum value was masked to emphasize the primary features.

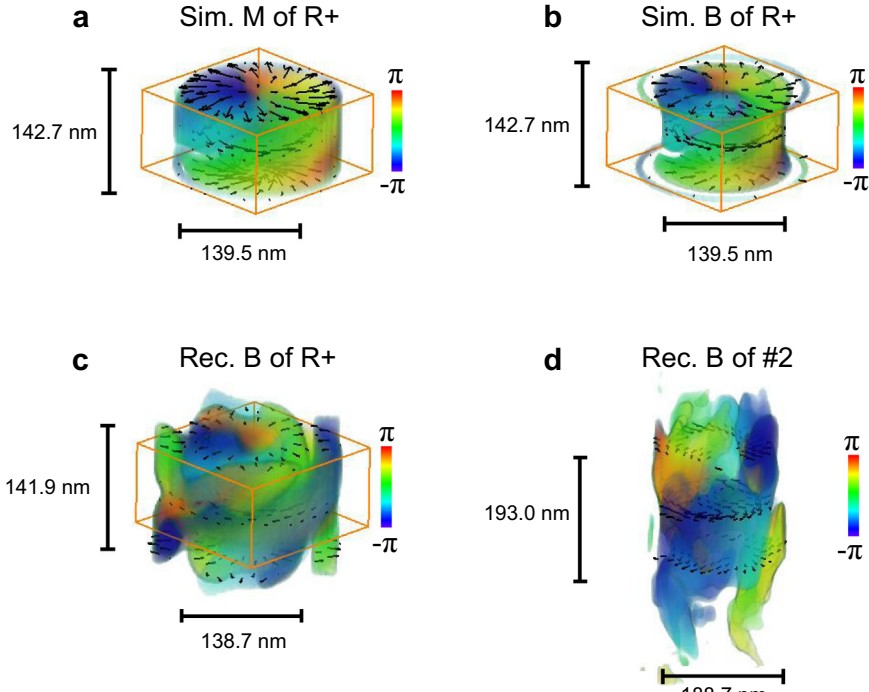

**Fig. 3 | The magnetic R+ configuration of bubble #2. a** Distribution of in-plane magnetizations in the 3D feature of the R+ model. **b** In-plane induction components computed with the R+ model. **c** In-plane induction components reconstructed from tilted LTEM images simulated based on the R+ model. **d** In-plane induction components recovered from the experimental data of bubble #2. The box frames in (**a**–**c**) indicate the physical size of the R+ model. Only pixels with values larger than 20% of the maximum intensity are rendered in (**b**–**d**) to depress the artifacts of the regularization parameter and the missing wedge effect. The color denotes the orientation of the in-plane components.

components in bubble #2 at different cross-section planes are shown in Supplementary Fig. 13. The successful reproduction of bubble #2 confirms the reliability of the reconstruction approach for micromagnetic objects, including the surface states at the boundaries predicted by the theory[33]. During the data acquisition process, out-of-plane magnetizations, which exist mainly in the core of the bubbles along the electron beam direction at near zero tilting angles, are difficult to align, resulting in some additional in-plane projections, as displayed in Fig. 2b.

Due to several factors, such as the missing wedge effect, the regularization parameter $q_0$, or the error in the thickness measurement, may reduce the accuracy of the induction value calculated at each voxel. However, the relative strength and orientation of the reconstructed components are still appropriate for judging the chirality of each bubble (Fig. 3c, d). Investigations of the simulated and experimental data indicate that the polarity of the bubbles can be inferred from the reconstructed surface features, i.e., convergent or divergent spin arrangements, without calculating the $B_z$ component.

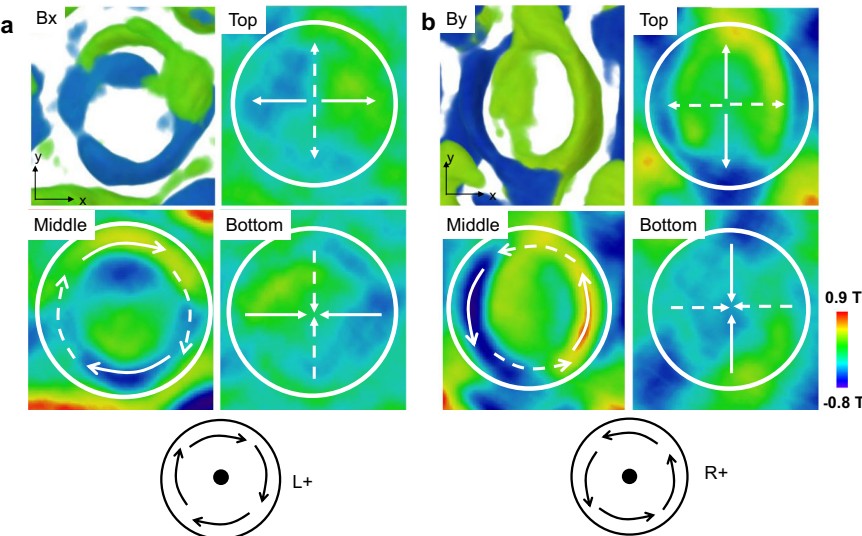

**Fig. 4 | The magnetic spin transition of bubble** #5. The $B_x$ (**a**) and $B_y$ (**b**) features in the top, middle, and bottom x-y sections. The solid arrows indicate the direction of the corresponding component, while the dashed arrows indicate the speculative direction of the orthogonal component based on prior knowledge of Bloch skyrmion bubbles. The color legend indicates the magnitude of induction components in the x or y direction, which corresponds to the magnetization between $-6.5 \times 10^5$–$7 \times 10^5$ A/m and well agrees with the measured value of $8 \times 10^5$ A/m[11].

**Table 1 | Comparison of a spin configuration of skyrmion bubbles between x-tilting and y-tilting**

| | X-tilting | | | | | | Y-tilting | | | | | |
|---|---|---|---|---|---|---|---|---|---|---|---|---|
| | Top | Middle | Bottom | p | w | Type | Top | Middle | Bottom | p | w | Type |
| #1 | D | CCW | C | + | - | R+ | D | CW | C | + | + | L+ |
| #2 | D | CCW | C | + | - | R+ | D | CCW | C | + | - | R+ |
| #3 | D | CCW | C | + | - | R+ | D | CW | C | + | + | L+ |
| #4 | D | CCW | C | + | - | R+ | D | CCW | C | + | - | R+ |
| #5 | D | CW | C | + | + | L+ | D | CCW | C | + | - | R+ |
| #7 | D | CCW | C | + | - | R+ | D | | C | + | | II |
| #8 | D | CCW | C | + | - | R+ | D | CCW | C | + | - | R+ |
| #9 | D | CW | C | + | + | L+ | D | | C | + | | II |
| #10 | D | CW | C | + | + | L+ | D | | C | + | | II |
| #11 | D | CW | C | + | + | L+ | D | | C | + | | II |
| #12 | D | CCW | C | + | - | R+ | C | | D | + | | II |
| #13 | D | CCW | C | + | - | R+ | D | CCW | C | + | - | R+ |

*D* divergent, *C* convergent, *CW* clockwise, *CCW* counterclockwise; *L+* left-handed, type I bubble with positive polarity, *R+* right-handed, type I bubble with positive polarity; *L-* left-handed, type I bubble with negative polarity, *R-* right-handed, type I bubble with negative polarity, *II* type-II bubble.

Furthermore, we note that only one set of x-tilting or y-tilting data, which exhibits radial or anti-radial nature on the surface of the bubbles, provides sufficient evidence for determining the polarity of the bubbles based on the prior analysis (Fig. 1a). Overall, we conclude that only one tilting experiment can confirm the bubble chirality: the surface features determine the polarity (*p*), while the in-plane components without tilting determine the vorticity (*w*) (Figs. 1b, 2b). This method was applied to explore different types of distorted bubbles in the two-tilting series, where only the $B_x$ or $B_y$ components were reliably reconstructed independently. For example, as shown in Fig. 4, the in-plane $B_x$ components of bubble #5 in the x-tilting experiment were convergent near the bottom surface, clockwise in the center, and divergent near the top surface. According to the models shown in Fig. 1, this bubble should be an L+ (*p+*, *w+*) type bubble. However, in the y-tilting situation, the in-plane $B_y$ components represent convergent, counterclockwise, and divergent characteristics, which belong to the R+ (*p+*, *w−*) type. Thus, bubble #5 reversed its spiral direction but maintained its polarity. Although the details of the

magnetic configurations within the bubble are not clear, the magnetic structure transformation between x-tilting and y-tilting was determined correctly.

The magnetic features of the bubbles marked in Fig. 2a were inspected further with the above approach and are summarized in Table 1. The corresponding 3D $B_x$ or $B_y$ components of all bubbles are shown in Supplementary Information. The table indicates that almost all types of bubbles retained their positive polarity (*p+*) in the x-and y-tilting data, regardless of how the vorticity (*w*) varied. Type I and type II bubbles have previously been found in the MnNiGa system[11,12,14,31,34–37]. It is well known that these bubbles are metastable and easily change their spin chirality or topological texture under external stimuli, which is not only demonstrated here but has also been observed in other literature[34,38]. Loudon et al. observed that the swirling direction of the in-plane induction in the skyrmion bubbles changed in the MnNiGa alloy, including the type transformation, according to the LTEM images[34]. However, in their experiments, the polarity of the bubbles was fixed by an in situ external field; therefore, the 3D structure was

not needed to determine the chirality of the bubbles. Thus, they cannot determine the variation in the chirality of the bubbles under field-free conditions. In our work, the bubbles were studied under almost field-free conditions in an LTEM (the residual field is less than 10 Oe). During the field precooling procedure, the out-of-plane component inside an individual bubble should be antiparallel to the direction of the magnetic field in the MnNiGa matrix to maintain its special topological characteristics. Thus, some external perturbations, such as rotating the specimen outside the LTEM or electron beam irradiation, may cause the bubble to change its in-plane magnetic texture but fail to disrupt the orientation of the out-of-plane component unless a strong external vertical field is applied. As a result, the bubbles alter their vorticity rather than their polarity, maintaining their existence in the MnNiGa bubble lattice. This characteristic of the DDI bubble has not been observed in previous in situ characterizations[34]. Thus, it appears that rearranging the out-of-plane components of the magnetic moments in the bubbles requires more energy than rearranging the in-plane components. Thus, a chirality oscillation in the bubble that transforms the swirling mode but maintains the polarity occurs in the centrosymmetric magnet MnNiGa.

## Discussion

The ideal specimen shape for TEM tomography is a needle-like or cylindrical object. These types of specimens are typically fabricated with the focus ion beam (FIB) method, since the needle specimen can be tilted at a large angle along the long axis, allowing high-angle data to be obtained and eliminating the missing wedge effect. However, in this case, the magnetic field reconstruction requires two image sequences tilted around two orthogonal axes. Therefore, the available angles are restricted to a narrower range when the needle-like sample is tilted around the shorter axis. Moreover, the shortest board determines the actual accessible angle range, which is similar to the conventional TEM sample used in our experiments. Because the missing wedge effect elongates the structure along the z direction, the spatial resolution in this direction is worse than that in the x-y plane. In addition, a nonzero regularization parameter $q_0$ results in the attenuation of primary structures, impeding quantitative calculations with the reconstructed data. However, in actual situations, the relative strength of the inductions is sufficient to determine the vector orientation and chirality of the bubble if a proper threshold is used to assess the primary features.

The flipping mechanism of the bubbles remains an open question. In the experiments, some bubbles varied their features, including their shape and swirling direction, under beam showering. Most bubbles remain stable under a constant beam current; however, changes in the brightness can result in some bubbles randomly flipping (see Supplementary Information). The beam can induce local heating, which may be the reason for the observed spin reorientation[32]. Therefore, after a curing period, a region containing stable bubbles was selected to acquire the x-tilting data. When the specimen was removed from the LTEM and rotated to acquire the y-tilting characterization data, the bubbles experienced a substantial change in brightness and reached a new balanced state when they reentered the electron beam of the LTEM. Therefore, some bubbles had different vorticities than in the first x-tilting observation.

Skyrmion bubbles (DDI skyrmions) were a core component in commercial nonvolatile memory devices in the last century; however, they have lost this distinction with the development of hard disk techniques. Nonetheless, the small size of individual bubbles is still an attractive factor for high-density memory applications due to the topological property of the magnetic unit. Because their vorticity can be modified by electron beams, skyrmion bubbles can be used in novel high-density/high-speed memory applications. Considering the concept of racetrack memory, which was designed for DMI skyrmions[39], a similar scheme for DDI skyrmions is shown in Fig. 5. A sequence of bubbles carried by a direct current (DC), with the same chirality and polarity,

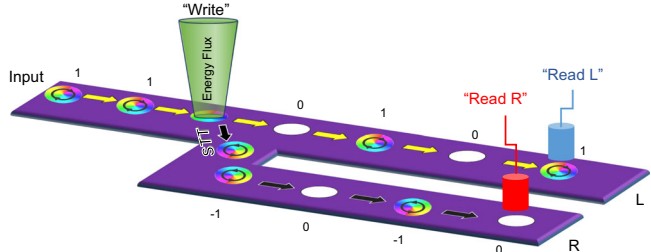

**Fig. 5 | A schematic diagram of a racetrack memory method using the DDI skyrmion.** The bubbles with uniform chirality move under an electonic beam which can change the in-plane swirling of those bubbles. Then STT effect seperates those bubbles to diffferent sides of the track and two sensors may read the corresponding spin texture as the desired signal information.

flows under a fine electron probe which is easily focused to several nanometers. A pulse signal controls the emission of the electron probe, varying the vorticity of the bubble passing through the electron beam while maintaining the polarity of the bubble. Due to the spin-transfer torque (STT) mechanism or the Magnus effect, bubbles with different vorticities are automatically separated into different branches downstream. Two sensors in each line work synchronously in this complementary device. An important point is that the polarity and vorticity are decoupled in DDI bubbles. If the polarity is also manipulated, one DDI skyrmion can demonstrate a new four-bit memory: two bits from the polarity and two bits from the vorticity. DMI skyrmions are believed to move easily under DC currents, while the DDI bubbles prefer to reverse their vorticity because of local pinning effects[13]. In contrast, the DMI skyrmion needs a vertical field to reverse its whirling axis and modify its spiral orientation because spin-orbital coupling binds the polarity and vorticity in DMI systems, preventing independent manipulations.

In summary, the 3D magnetic configurations of skyrmion bubbles were revealed by a vector field tomography approach in an LTEM. The surface magnetic twisting of the bubbles was confirmed by the reconstructed features. The vorticity and polarity of the bubbles were both determined, and the types of bubbles were directly identified. Furthermore, we demonstrate that the type of bubble can be estimated from only one set of tilted data, simplifying the experimental procedure. In addition, the transformation of the bubbles was clarified by the retrieved magnetic configuration, and we found that the variation in the spins was limited to the vorticity because the bubble spontaneously maintains its polarity to preserve its singular state in the matrix. The stable polarity of skyrmion bubbles is useful for potential applications in future devices.

## Methods

### Sample fabrication

The thin plate for the LTEM characterization was cut from a bulk alloy and prepared by mechanical polishing, followed by Ar+ ion milling. The specific field cooling (FC) manipulation was performed with a physical property measurement system (PPMS). The sample was fixed in the puck and placed into the cavity. The sample was heated to 380 K, which is higher than the Curie temperature ($T_C$ ~345 K). Then, a small magnetic field of 500 Oe was applied, and the sample was gradually cooled to 300 K, at which point the field was turned off.

### LTEM characterization

The skyrmion bubbles were investigated with a JEOL-dedicated Lorentz TEM (JEOL2100F, residual magnetic field below 10 Oe) equipped with a high tilted holder and a JEOL Cs-corrected ARM200F appended Lorentz mode. No magnetic field was applied when performing the experiment. The coordinates were determined in the laboratory frame, with the direction of the electron beam defined as the z-axis and the perpendicular plane defined as the x-y plane. The thickness of the

observation region was between 150–180 nm and was estimated based on the ratio between the plasmon and zero loss peaks in the electron energy loss spectrum (EELS). The out-of-plane component of the bubble, or the easy [001]-axis of MnNiGa, was aligned as close as possible to the electron beam at the initial position (tilt angle $\theta = 0$). The specimen was tilted around the x-axis first in the LTEM experiments, then manually rotated 90° outside the electron microscope and tilted around the y-axis in the LTEM. The x-tilting ranged from −54.8° to 52.1°, and the y-tilting ranged from −57.2° to 56.2°, with an interval of 1–3°. The uneven interval prevents images containing strong diffraction contrast and ensures correct phase retrieval in the TIE. Three images, namely, an overfocused image, an in-focus image, and under focused image, were acquired for each $\theta$.

## 3D reconstruction

The images acquired at each tilting angle $\theta$, including the under-focused, in-focus, and overfocused images, were processed by the TIE approach to obtain the corresponding phase images after an image registration process that corrected the shape distortions caused by large changes in the focus among the images. In the TIE approach, the phase image was calculated based on the differential of images acquired at different focuses,

$$\varphi(x,y,z_0) = -k\nabla_{x,y}^{-2}\left\{\nabla_{x,y}\left[\frac{\nabla_{x,y}\nabla_{x,y}^{-2}\left(\frac{\partial I(x,y,z)}{\partial z}\right)}{I(x,y,z_0)}\right]\right\} \quad (1)$$

where $\nabla_{x,y}^{-2}$ is the inverse Laplacian operator. With a Fourier transform (FT), the inverse Laplacian operator shown in Eq. 1 can be formulated as $\nabla_{x,y}^{-2} \rightarrow \mathcal{F}^{-1}\left\{\frac{-\mathcal{F}[\bullet]}{|\mathbf{q}|^2}\right\}$, where $\mathbf{q}$ is a vector in the x-y plane of the frequency domain. To prevent singularities when $\mathbf{q} = 0$, a regularization parameter $q_0$ was introduced, $\nabla_{x,y}^{-2} \rightarrow \mathcal{F}^{-1}\left\{\frac{-\mathcal{F}[\bullet]|\mathbf{q}|^2}{(|\mathbf{q}|^2 + q_0^2)^2}\right\}$. In this work, $q_0$ was $5 \times 10^{-3}$ nm$^{-1}$ to compromise between efficiently reducing the low-frequency noise and eliminating the artificial signal[14,15]. The generated phase image stack was automatically aligned with a bandpass filter by the "Image Alignment" plug-in function in the Gatan Microscopy Suite (GMS). A preferable method is to select a region containing bubbles with stronger contrast as the reference to improve the alignment quality. The well-aligned phase images were superimposed into one image, and its FT provided a good indication of the tilting axis, showing that the extending direction of the diffused background was along the tilt axis[40]. The maps of the induction component parallel to the tilt axis were computed according to the differential phase in the orthogonal direction; for example, Bx map was computed from the derivative of the phase in y direction. The resulting maps were input into the W-SIRT[41] program in the GMS to reconstruct the 3D $B_x$ and $B_y$ arrays.

The $B_x$ and $B_y$ arrays were manually registered and reshaped to the same size. Because $\nabla \cdot \mathbf{B}(\mathbf{r}) = 0$, it is easy to determine that $\mathbf{q} \cdot \mathbf{B}(\mathbf{q}) = 0$ by an FT, where $\mathbf{B}(\mathbf{q})$ is the Fourier coefficient of $\mathbf{B}(\mathbf{r})$ and $\mathbf{q}$ is the coordinate in reciprocal space. Thus, $B_z$ can be computed in the frequency domain and transformed back to real space.

$$q_x B_x(\mathbf{q}) + q_y B_y(\mathbf{q}) + q_z B_z(\mathbf{q}) = 0 \quad (2)$$

$$B_z(\mathbf{q}) = -\frac{q_x B_x(\mathbf{q}) + q_y B_y(\mathbf{q})}{q_z} \quad (3)$$

$$B_z(\mathbf{r}) = \mathcal{F}^{-1}\left[B_z(\mathbf{q})\right] \quad (4)$$

It should be noted that when the frequency is zero in Eq. 3, a direct current component must be weighted manually to satisfy the physical boundary conditions, such as $B_z$ in the far region, which must be zero. The postprocessing and visualization of the data were accomplished with Avizo®.

## Data availability

All the data that support the findings of this study are presented in the paper and are available from the corresponding author upon reasonable request.

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

## Acknowledgements

The authors thank Prof. Ying Zhang for her assistance with the TEM characterization. This work was supported by the National Natural Science Foundation of China (No. 11874410 and 12274321), the National Key R&D Program of China (No. 2021YFB3501402), and the Strategic Priority Research Program of the Chinese Academy of Sciences (No. XDB33000000).

## Author contributions

Y.Y. and W.W. supervised the project; Y.Y. and B.D. proposed the idea and designed the experiments; H.L. synthesized the MnNiGa polycrystal; B.D. and J.L. prepared the lamella and performed the Lorentz-TEM measurements; Y.Y. analyzed the experimental data and completed the reconstruction; and Y.Y. and D.B. wrote the manuscript after discussing the data with the other authors.

## Competing interests

The authors declare no competing interests.
