## [Peer Review File · Nature Communications]

Reviewers' Comments:

Reviewer #1:

Remarks to the Author:

This manuscript reports on dynamic chirality flips of skyrmion bubbles found in a thin plate cut from a bulk sample of $(\text{Mn}(1-x)\text{Ni}x)_65\text{Ga}_{35}$ ($x = 0.45$). Magnetic skyrmions have been found in chiral non-centrosymmetric crystals such as MnSi, FeGe, Cu_2OSeO_3 , CoZnMn , GaV_4S_8 etc. MnNiGa is centrosymmetric and hence cannot host conventional magnetic skyrmions. The authors have undertaken Lorentz transmission electron microscopy studies to investigate the structure of the magnetic objects found in MnNiGa . Using vector field tomography they have demonstrated that the objects are in fact skyrmion bubbles. They also demonstrate that these skyrmion bubbles can be observed to flip their chirality and transform their vorticity whilst retaining their positive polarity.

Unfortunately, nearly all of the conclusions they report in this paper have been reported, and published, before, and in papers not referenced by the authors. Loudon et al (*Advanced Materials* 31 (16) 1806598 2019) have previously reported a very similar study using Lorentz TEM on the same material and found both type I and type II bubbles and that the previously reported biskyrmions were in fact type II bubbles. They also demonstrated the internal structure of these objects using both x-ray holography and Lorentz TEM. Micromagnetic simulations and quantitative transport of intensity equation were used to determine the magnetisation of the objects. Figure 4 of their paper also shows the transformation, chirality flips, of individual bubbles in MnNiGa . I note that Loudon et al include a reference in their paper to an arXiv paper published by Y. Yao et al and including four of the manuscript authors. I am surprised that the authors of this manuscript have not found it worthwhile referencing the paper by Loudon et al.

This manuscript would be greatly improved by the authors referencing previous relevant studies and restricting their manuscript to the novel aspects which have not been published. The use of vector field tomography via LTEM is a wonderful demonstration and does have advantages over the earlier study by Loudon. However, the authors are not able to explain these advantages without referencing the earlier studies. I also note a later paper published by Turnbull et al (*ACS NANO* 15 (1) 387 2021) reporting on tilting x-ray holography of magnetic bubbles in the same material, which is a technique making progress towards vector tomography. The authors should also consider the arXiv paper published by Nakajima "Two types of magnetic bubbles in MnNiGa observed via Lorentz microscopy" arXiv:2110.15507 (2021) and also S.L. Zuo et al., *Nanoscale* 10, 2260 (2018).

I believe this manuscript is not worthy of publication in *Nature Communications* and should be rejected. The authors should in future take more care to adequately reference papers published by others in the field and to give suitable credit to them. The current manuscript certainly does contain novel and interesting results but is not worthy of publication in its present form.

Reviewer #2:

Remarks to the Author:

Review of manuscript NCOMMS-21-41870 entitled „The dynamic chirality flips of Skyrmion bubbles“ by Yuan Yao, Bei Ding, Jinjing Liang, Hang Li, Xi Shen, Richeng Yu, Wenhong Wang

Yuan Yao, Bei Ding and coworkers report on the tomographic 3D reconstruction of skyrmion bubbles in Mn-Ni-Ga from tilt series of Lorentz transmission electron micrographs. The goal of these investigations is to determine the 3D magnetic spin structure of different types of bubbles in terms of chirality, polarity and topology that is not possible from single L-TEM images. As a result, they found that bubbles switched their chirality (between reconstructions of two different Cartesian B-field components) but still keep the polarity to remain the singularity of the bubbles within the material.

I find the approach very appealing and suited to be published in *Nature Communications*. However, the paper in its current stage lacks from numerous experimental uncertainties, logical

and theoretical inconsistencies, and physical flaws and hence I can only recommend it for publication after a careful and major revision considering the points below.

The major points of criticism which have to be addressed are the following:

1. The magnetic states of the Skyrmion bubbles seem to be very unstable (metastable) at RT and therefore susceptible to be modified by small magnetic fields (compare e.g. in Fig.2a left and right column). The authors observe this problem by comparing the 3D induction maps of B_x and B_y (e.g. Fig.4 and Table 1). Since B_x and B_y belong only to the same magnetic state in bubble #2 only, they could only be combined in this case to obtain the full 3D vector-field by solving $\text{div}B=0$. Could it be that the residual magnetic field of the Lorentz lens (how large?) has changed the magnetic states of the bubbles when the sample was retracted and inserted in the TEM for in-plane rotation? Even worse, did the authors have detected some changes within tilt series acquisition, i.e. when going to higher tilt angles due to the changed orientation of the residual field with respect to the specimen? To avoid this unintended flipping of the magnetic states, it would have been more favorable to cool the sample during acquisition or to use an in-plane rotation holder.

2. The four magnetic states of the type I bubbles were computed by micromagnetic simulations as shown in Fig. 1. The Neel twisting at their surfaces leads to magnetic charges, i.e. a high contribution of H , but a low one of B . Since, L-TEM is only sensitive to B not M , it is worthwhile to know how the simulated B -field looks like especially at the surface. In this context, it is also interesting to know how extended in z -direction these Neel twists are and whether they can be resolved by VFET.

3. The experimental tilt range is quite low: ca. $\pm 55^\circ$ instead of $\pm 90^\circ$. Therefore the tomograms suffer from a large missing wedge, not discussed in the manuscript at all. I suppose, one problem is the slab geometry of the (FIB?) lamella, which is disadvantageous at high tilts, when new object information appears in the field-of-view and the projected thickness becomes thicker and thicker. How the authors tackled this issue especially when using a SIRT algorithm for reconstruction, where the reconstructed volume must be limited? Why the authors have not tried to prepare a needle-shaped sample much more suitable for tomography?

4. To judge the fidelity of the tomograms, it is also important to know how the applied focus of $300 \mu\text{m}$ determines the lateral resolution of the L-TEM images described by the spatial envelope. Also the regularization (not regularity as written in the paper) parameter q_0 (line 272) determines on a certain way the fidelity of the 3D data. In best case, the authors should check by a simulated tilt series using same parameters (tilt range, defocus, q_0), whether the tomogram reveals the original magnetic structure?

5. With respect to the resolution limitations mentioned in points 3 and 4, the observation of a very localized Bloch line in Figs. 3c,d is really questionable. The authors should check the reliability of this feature.

6. Why the authors do not provide the reconstructed 3D B-fields in Tesla. E.g., in Fig.4 the values range from $\pm 10^6$ (A/m?). They must not be denoted M_x and M_y , because these are B-fields.

Minor comments

1. Can the authors give more information/data about the experiment and data treatment (at least in the Suppl. Mat.), such as

- TEM lamella preparation (by FIB?)
- Field cooling procedure
- Lorentz TEM tilt series (to be shown in supplement?)
- TIE reconstruction (influence of regularization)
- phase tilt series (to be shown in supplement?)
- Image alignment (line 275)
- W-SIRT parameters (influence of regularization)

2. Mathematics:

Could the authors more clearly describe the mathematics of TIE?

- As shown by Lubk et al. (PRL 111, 173902 (2013)) Eq. 1 is only valid, if a) the object is a pure phase object, or b) the current density is conservative. What is valid here?
- What is the meaning of symbol ∇^{-2} ? Isn't it the inversion of Laplacian, $1/q^2$ in Fourier space to be inserted in line 271 on the LHS instead ∇_{xy} ?
- Inconsistencies of notation, in Eq. (1) reciprocal vector is q , whereas in Eq. (2) reciprocal vector is k

3. English writing and wording.

- there are numerous typos and sometimes inadequate choice of words, e.g.

line 20: "magnetic moment" is misleading here, line 40 "need", line 48 "turns into" not appropriate in this context, line 64 "charity", line 116 "configuraiton", line 232, 238 "configure", line 263 "3D structure reconstruction" not the right term

The text needs very much polishing in general.

4. Fig.1b: Why there is a threshold in the simulated induction maps (black mask)?

5. Line 273: Please provide the value of q_0 in "1/nm" not pixels.

Reviewer #1 (Remarks to the Author):

This manuscript reports on dynamic chirality flips of skyrmion bubbles found in a thin plate cut from a bulk sample of $(\text{Mn}(1-x)\text{Ni}_x)\text{Ga}_{0.35}$ ($x = 0.45$). Magnetic skyrmions have been found in chiral non-centrosymmetric crystals such as MnSi, FeGe, Cu_2OSeO_3 , CoZnMn, GaV4S8 etc. MnNiGa is centrosymmetric and hence cannot host conventional magnetic skyrmions. The authors have undertaken Lorentz transmission electron microscopy studies to investigate the structure of the magnetic objects found in MnNiGa. Using vector field tomography they have demonstrated that the objects are in fact skyrmion bubbles. They also demonstrate that these skyrmion bubbles can be observed to flip their chirality and transform their vorticity whilst retaining their positive polarity.

Unfortunately, nearly all of the conclusions they report in this paper have been reported, and published, before, and in papers not referenced by the authors. Loudon et al (Advanced Materials 31 (16) 1806598 2019) have previously reported a very similar study using Lorentz TEM on the same material and found both type I and type II bubbles and that the previously reported biskyrmions were in fact type II bubbles. They also demonstrated the internal structure of these objects using both x-ray holography and Lorentz TEM. Micromagnetic simulations and quantitative transport of intensity equation were used to determine the magnetisation of the objects. Figure 4 of their paper also shows the transformation, chirality flips, of individual bubbles in MnNiGa. I note that Loudon et al include a reference in their paper to an arXiv paper published by Y. Yao et al and including four of the manuscript authors. I am surprised that the authors of this manuscript have not found it worthwhile referencing the paper by Loudon et al.

This manuscript would be greatly improved by the authors referencing previous relevant studies and restricting their manuscript to the novel aspects which have not been published. The use of vector field tomography via LTEM is a wonderful demonstration and does have advantages over the earlier study by Loudon. However, the authors are not able to explain these advantages without referencing the earlier studies. I also note a later paper published by Turnbull et al (ACS NANO 15 (1) 387 2021) reporting on tilting x-ray holography of magnetic bubbles in the same material, which is a technique making progress towards vector tomography. The authors should also consider the arXiv paper published by Nakajima “Two types of magnetic bubbles in MnNiGa observed via Lorentz microscopy” arXiv:2110.15507 (2021) and also S.L. Zuo et al., Nanoscale 10, 2260 (2018).

I believe this manuscript is not worthy of publication in Nature Communications and should be rejected. The authors should in future take more care to adequately reference papers published by

others in the field and to give suitable credit to them. The current manuscript certainly does contain novel and interesting results but is not worthy of publication in its present form.

R: We totally refuse the comments from the referee. We are familiar with Loudon's paper in *Advanced Materials* (31 (16) 1806598 2019) which also cited the arXiv version of our previous paper (the published version is *Applied Physics Letters*, 114, 102404, 2019). Both his AM paper and our previous APL paper claimed that the biskrymions should be interpreted as the Type II bubbles. We appreciate the experimental design of combining the X-ray holography and LTEM imaging techniques to illuminate the magnetic detail of Type I and Type II bubbles. They also found that the structure changes from Type I to Type II bubbles under ***applying external magnetic field***.

1. However, Loudon's work used X-ray to characterize the out-plane component (B_z) of the Type I bubbles but LTEM to observe the in-plane component (B_x - B_y) of the bubbles. Obviously, the combination of these two sets of data cannot confirm the chirality of ***individual*** bubble, whatever the Type I or Type II because the in-plane components of the moments are lost in X-ray experiments and the polarity of each bubble disappears in LTEM images. Although the polarity (B_z) may be confirmed in X-ray investigation, the actual chirality of bubbles in LTEM could not be directly determined from their swirling style of in-plane components unless they pre-specified the B_z direction of each bubble in LTEM (***of cause they cannot***), which is indicated clearly in fig. 1 of our submitted manuscript. Indeed based on Loudon's data, it is difficult to build the real 3D magnetic structure of bubbles and hard to claim that "***demonstrated the internal structure of these objects***" as referee said. Loudon et al employed micromagnetic simulation, a conventional method, to illustrate the 3D structure of the bubbles in their paper but it does not mean that they ***really observed*** the entire configurations in the 3D space. Compared with them, we did reconstruct the entire structure of bubbles in MnNiGa ***including the internal and surface moment orientation*** and this is exactly the novelty of our manuscript. We also indicated that based on the reconstructed 3D feature of the B_x ***or*** B_y component, the chirality of bubbles can be clarified qualitatively, which is another important contribution of our work. It seems like a natural anticipation from the simulation but cannot be revealed by Loudon's work or other literature listed by the referee. The referee's conclusion that Loudon et al have reconstructed the magnetic structure of the bubbles in MnNiGa is a ***misleading statement***.
2. The referee said that the fig. 4 in Loudon's AM paper displayed the helical reversal of the bubbles and he concluded Loudon et al also confirmed the chirality changes as our reported. But referee forgets that ***there was an external out-plane field of 233 mT applied on the sample!*** (First sentence in second-to-last paragraph, page 4 of Loudon's paper.) Based on this condition, one can suppose that the B_z of bubbles were fixed and the reversal of the in-plane components implied the change of the chirality. However, without this assumption

or in a field free situation as described in our experiments, Bx-By changes could not infer whether the polarity varies simultaneously. If both vorticity (what can be seen in Loudon's work) and polarity (what cannot be seen in Loudon's work) change, the chirality maintains unchanged! Clearly, distinguishing what happens cannot be achieved by the traditional LTEM images in Loudon's paper or other's. The referee may skip the fig1 in our manuscript which demonstrates the importance of the real 3D exploration for spiral magnetic structures. We denoted this point and realized this discrimination in MnNiGa system for the more general case, which was impossible to be accomplished in Loudon's AM paper or other literature mentioned by the referee since the features in several tilted LTEM images cannot supply enough information to determine the detail of the magnetic structures. Actually, those papers did not show the conclusion that they really detected the chirality of studied bubbles at a field free condition.

3. Turnbull's paper (ACS NANO 15 (1) 387 2021, from the same research group with Loudon) described the progress in detecting the Type I and Type II bubbles by X-ray holography and an approach using two projection images at different angles to judge the type of bubbles. However, the vorticity of the in-plane components cannot be explored because X-ray is only sensitive to the components parallel the propagating direction. Tilting the sample also impossibly supplies the rotation manner of the in-plane features. This method cannot even demonstrate the swirling style of the bubble, let along the chirality! The highlights of our work are to draw the 3D configuration of the magnetic bubbles and clarify their topological characteristics, but are not to prove the existence of type-I or type-II bubbles in MnNiGa alloy. The arXiv paper of Nakajima 2110.15507 (2021) found there may be two kinds of Type II bubbles in MnNiGa, characterized by LTEM images. Zuo's paper studied the so-called "biskyrmions" in MnNiGaY. Both of two literatures did not mention the 3D structure or the chirality of the magnetic bubbles. We cannot understand why the referee mentioned these literatures as the evidence to debase the novelty of our work.

We respect Loudon's beautiful works and other related outstanding researches. Since we focused on the 3D structure recovery of the magnetic structure, their papers were not referenced in our manuscript. We can give more comparison with their reports in our work but cannot accept referee's *misleading and biased comments*. Some literature have been appended in the revised manuscript. Thus, we strongly appeal to editor to reconsider our modified manuscript.

Reviewer #2 (Remarks to the Author):

Review of manuscript NCOMMS-21-41870 entitled „The dynamic chirality flips of Skyrmion bubbles“ by Yuan Yao, Bei Ding, Jinjing Liang, Hang Li, Xi Shen, Richeng Yu, Wenhong Wang

Yuan Yao, Bei Ding and coworkers report on the tomographic 3D reconstruction of skyrmion bubbles in Mn-Ni-Ga from tilt series of Lorentz transmission electron micrographs. The goal of these investigations is to determine the 3D magnetic spin structure of different types of bubbles in terms of chirality, polarity and topology that is not possible from single L-TEM images. As a result, they found that bubbles switched their chirality (between reconstructions of two different Cartesian B-field components) but still keep the polarity to remain the singularity of the bubbles within the material.

I find the approach very appealing and suited to be published in Nature Communications. However, the paper in its current stage lacks from numerous experimental uncertainties, logical and theoretical inconsistencies, and physical flaws and hence I can only recommend it for publication after a careful and major revision considering the points below.

The major points of criticism which have to be addressed are the following:

1. The magnetic states of the Skyrmion bubbles seem to be very unstable (metastable) at RT and therefore susceptible to be modified by small magnetic fields (compare e.g. in Fig.2a left and right column). The authors observe this problem by comparing the 3D induction maps of B_x and B_y (e.g. Fig.4 and Table 1). Since B_x and B_y belong only to the same magnetic state in bubble #2 only, they could only be combined in this case to obtain the full 3D vector-field by solving $\text{div}B=0$. Could it be that the residual magnetic field of the Lorentz lens (how large?) has changed the magnetic states of the bubbles when the sample was retracted and inserted in the TEM for in-plane rotation? Even worse, did the authors has detected some changes within tilt series acquisition, i.e. when going to higher tilt angles due to the changed orientation of the residual field with respect to the specimen? To avoid this unintended flipping of the magnetic states, it would have been more favorable to cool the sample during acquisition or to use an in-plane rotation holder.

R: The residual magnetic field in LTEM JEOL 2100F-LM is below 10 Oe since it is a special designed instrument for the magnetic object characterization. So the residual field should not be a reasonable influence to the observed feature changes of the bubbles. **The appearance of the bubbles varied little during the sample tilting, which has been testified by the images before and after tilting (fig. s14 in supplemental information or fig. r1 here).** Indeed, the reconstruction of each B_x or B_y may include the tolerance of these possible distortion and strictly speaking the results were semi-quantitative to illuminate the moments behavior within the bubbles. We also avoid the quantificational description for the magnetic moment because of the regularization

parameter, the missing-wedge effect and the uncertain in B_z calculation. Fortunately, these semi-quantitative features containing some details can supply enough information to investigate the type or the chirality of the bubbles. Unfortunately, the features of some bubbles changed obviously when sample rotated 90 degrees for acquisition in another tilting axis. The environment changes during operation **MAY** be considered but we have to say that the exact reason is not clear right now. One confirmation is that the configuration of the bubbles in MnNiGa is not stable. Cooling down the sample is a good suggestion to avoid some unknown effects but the tomography experiment needs a high-tilting holder, which excludes the possibility of the cooling holders in present study. In-plane 90 degrees rotation holder is not a general available apparatus for the tomography exploration now, but it is a valuable direction to be developed.

Fig. rf The features of bubbles did not change during the x-tilt operation.

2. The four magnetic states of the type I bubbles were computed by micromagnetic simulations as shown in Fig. 1. The Neel twisting at their surfaces leads to magnetic charges, i.e. a high contribution of H , but a low one of B . Since, L-TEM is only sensitive to B not M , it is worthwhile to know how the simulated B -field looks like especially at the surface. In this context, it is also interesting to know how extended in z -direction these Neel twists are and whether they can be resolved by VFET.

R: Yes, the LTEM only detect the magnetic induction B , not the magnetization M or magnetic field strength H . Following the suggestion, we computed the induction B in 3D space from the type I magnetic bubble with $R+$ configuration and verify the similar Neél features existing in both surfaces of the bubble, as the surface features in M cylinder. Of cause the size of B cylinder is

slightly larger than the M configuration and there are some differences resulted from the demagnetization in the simulation. These contents are added in the modified manuscript (fig. 3 and related discussion in manuscript, and fig. r2 here).

Fig. r2 The magnetic R+ configuration of 2# bubble. **a** Distribution of in-plane magnetizations in 3D feature of the R+ model. **b** In-plane induction components computed from R+ model. **c** In-plane induction components reconstructed from the tilted LTEM images simulated based on R+ model. **d** In-plane induction components recovered from the experimental data of 2# bubble. The box frames in a)-c) indicate the physical size of R+ model. Only the pixels with values larger than 20% of the maximum intensity are rendered in b)-d) to depress the artifacts from regularization parameter and missing-wedge effect.

“Fig. 3a interprets the rendered orientation of in-plane magnetization components within simulated R+ model in three dimensions, demonstrating the surface convergent or divergent features, while Fig. 3b shows the calculated in-plane inductions. The induction field involving stray strength in Fig. 3b replicates the moment style of the R+ bubble. In order to testify the fidelity of the reconstruction, serial tilting LTEM images were simulated from R+ model with same experimental conditions to rebuild the induction shape with those processing parameters for experimental task. The product in Fig. 3c demonstrates the reliable recovery of the R+ bubble but bears some artifacts from the TIE parameters and missing-wedge effect (the weak extension along z direction owing to the limited tilting range in data acquisition). Fig. 3d outlines the in-plane induction retrieved for 2# bubble, which appears similar contour in Fig. 3c and confirms the R+ type.”

3. The experimental tilt range is quite low: ca. $\pm 55^\circ$ instead of $\pm 90^\circ$. Therefore the tomograms suffer from a large missing wedge, not discussed in the manuscript at all. I suppose, one problem is the slab geometry of the (FIB?) lamella, which is disadvantageous at high tilts, when new object information appears in the field-of-view and the projected thickness becomes thicker and thicker. How the authors tackled this issue especially when using a SIRT algorithm for reconstruction, where the reconstructed volume must be limited? Why the authors have not tried to prepare a needle-shaped sample much more suitable for tomography?

R: Yes, the limited tilting range is the intrinsic shortage of present tomography in TEM. Frankly speaking, the missing-wedge is the problem difficult to be solved in TEM tomography because of the narrow space between the pole pieces. FIB prepared needle is more suitable for scalar field tomography since it can be tilted along the long axis of the cylinder to increase the tilt range. But for vector field reconstruction, the sample should be tilted along the short axis for second component reconstruction and eventually the tilt angle is also limited by its cylinder shape. At last the available common range for two tilting axes is not as large as expected. A wider region can be investigated in Ar⁺ ion milling sample (not FIB prepared sample) used in our experiment, which favors the discovery of the bubbles. The prolonged features along z-axis are observed in our reconstructed 3D features of B_x or B_y component. It deteriorates the resolution in z direction and carried out worse spatial resolution for B_z component. We depressed the data below 20% of maximum value of the corresponding components in our work to remove the missing-wedge effect. Although it is not a good choice for accuracy, it is an effective approach to remove the attenuate extension part and to recognize the contour of magnetic microstructure. Both simulations and practices verified the validation of this method. Conquering this difficulty needs more technical improvement in data acquisition or data processing but it is out the scope of present work. We add some discussions about this issue in the modified manuscript and emphasize that the retrieved moment figuration is a semi-quantitative result in order to avoid misleading the readers. It should be denoted that the relative strength of B vector, although less spatial resolution and data accuracy, could believably depict the chirality of the bubbles. They are contained in the modified version.

4. To judge the fidelity of the tomograms, it is also important to know how the applied focus of 300 μm determines the lateral resolution of the L-TEM images described by the spatial envelope. Also the regularization (not regularity as written in the paper) parameter q_0 (line 272) determines on a certain way the fidelity of the 3D data. In best case, the authors should check by a simulated tilt series using same parameters (tilt range, defocus, q_0), whether the tomogram reveals the original magnetic structure?

R: The lateral spatial resolution of the LTEM under large defocus (such as $300\ \mu\text{m}$) is about $5\ \text{nm}$, a coarse estimation. We try to use the simulated images with different tilting to reconstruct the entire virtue bubble to testify the validation of the method and parameters. We also tilted the simulated B configuration (R+) from -50° to 50° with 2° internal along tow orthogonal axes, generated the Fresnel images at each angle, retrieve each phase image with regularization parameter $q_0=5\times 10^{-3}\text{nm}^{-1}$ and reconstructed the entire induction matrix, as same as the experimental pipeline. Fig. s5 in supplemental information or fig. r3 here displays some simulated Fresnel images for different angles. The virtual operation demonstrated the parameters in experiment can ensure the recovery of the magnetic induction, while $q_0=5\times 10^{-3}\text{nm}^{-1}$ may enhance the extra artificial structure in the simulated structure than in the experimental data because of the low noise in the simulated images. This information has been appended in the revised manuscript.

Fig. r3 The simulated LTEM images for a R+ skyrmion. a) and b) under-focus and over-focus without tilting; c) and d) under-focus and over-focus with tilting -30° around x axis.

5. With respect to the resolution limitations mentioned in points 3 and 4, the observation of a very localized Bloch line in Figs. 3c,d is really questionable. The authors should check the reliability of this feature.

R: We understand the concern from the reviewer. The approximate location of the Bloch line could be estimated by the orientation change of the inductions although the resolution is not perfect for present results. We removed that statement in the manuscript to avoid the confusing presentation.

6. Why the authors do not provide the reconstructed 3D B-fields in Tesla. E.g., in Fig.4 the values range from $\pm 10^6$ (A/m?). They must not be denoted M_x and M_y , because these are B-fields.

R: Actually, the initial motivation of this study is to obtain the qualitative field of the bubble. But we found that the TIE method employed here can unavoidably distort the absolute value of the retrieved field because of the regularization parameter q_0 though the relative value may be credible. Therefore, we abandoned to evaluate the real value of B or M. The used calculated value at each pixel from the reconstruction algorithm is to illustrate the relative changes of B components. It is correct true that the LTEM investigates the B-field, not M. We have modified the figures and added more pictures about the computed B field and recovered B field retrieved from the simulated tilting images. Please see Fig. 3 and related discussion in manuscript, or fig. r2 here.

Minor comments

1. Can the authors give more information/data about the experiment and data treatment (at least in the Suppl. Mat.), such as

- TEM lamella preparation (by FIB?)

R: The Ar⁺ ion milling process was added in the Method section.

- Field cooling procedure

R: The specific FC manipulation was performed in the Physical Property Measurement System (PPMS). The sample was fixed in the puck and put into the cavity. It was heated to 380 K, which is higher than the Curie temperature ($T_C \sim 345$ K). Then a small magnetic field of 500 Oe was applied and the sample was cooled gradually to 300 K at which the field turned off. It was added in Method section.

- Lorentz TEM tilt series (to be shown in supplement?)

R: The some images in the experimental tilting series were shown in the supplement file.

- TIE reconstruction (influence of regularization)

R: The affect from the regularization q_0 on retrieved phase images are shown in supplement for the experiment data.

Fig. r4 The in-plane components of induction retrieved with a) $q_0=5\times 10^{-3} \text{ nm}^{-1}$ and b) $q_0=7.5\times 10^{-3} \text{ nm}^{-1}$, respectively.

- phase tilt series (to be shown in supplement?)

R: Due to limitations on space, some phase images from the tilting phase series are shown in Fig. s9 and s10 in supplement. We add the movie files of the phase stacks.

- Image alignment (line 275)

R: The phase images, such as for x-tilting, were formed to one stack file and aligned with the “Image Alignment” function in Gatan Microscopy Suite (GMS). A “bandpass filter” with default mode was employed to “automatic” register the images, followed by forward and backward corrections. After alignment, all phase images were merged into one image and that image were transformed to Fourier space. The prolonging diffuse background of the auto-correlation region (center part) in diffractogram indicated the tilt axis orientation, as shown in fig. r5. After the alignment, all phase images were rotated to make the tilting axis vertical or horizontal to the image boundary, facilitating the followed calculation of B_x or B_y components. It is denoted in the manuscript and some details are demonstrated in supplement.

Fig. r5 Diffractionogram of the merged phase image, where the dash denotes the orientation of tilt axis.

- W-SIRT parameters (influence of regularization)

R: The parameter in GUI of reconstruction plugin is added in the supplements. We think the referee wants to check whether different q_0 can make a significant change in the reconstructed 3D features. We added the investigation for reconstruction of the R+ model skyrmion in supplement to show this influence.

2. Mathematics:

Could the authors more clearly describe the mathematics of TIE?

- As shown by Lubk et al. (PRL 111, 173902 (2013)) Eq. 1 is only valid, if a) the object is a pure phase object, or b) the current density is conservative. What is valid here?

- What is the meaning of symbol $\nabla_{x,y}^{-2}$? Isn't it the inversion of Laplacian, $1/q^2$ in Fourier space to be inserted in line 271 on the LHS instead $\nabla_{x,y}^{-2}$?

- Inconsistencies of notation, in Eq. (1) reciprocal vector is q , whereas in Eq. (2) reciprocal vector is k

R: The principle of TIE is added in Supplement file. TIE is used to recover the phase of a wave, exactly the wave exiting the object in TEM experiments. Interpreting the physical meaning of the phase is not the responsibility of TIE, so actually it needn't any assumption for the wave characteristics. Eq. 1 only requires a conservation of Poynting current, described by the original Paganin's paper (D. Paganin and K. A. Nugent, Noninterferometric Phase Imaging with Partially Coherent Light, Phys. Rev. Lett., 80, 2586, 1998). Because the scattering ability of the specimen

is very weak so most features in images satisfy the conservation except the image boundaries which is out the interesting range in our studies. Eq. 1 is valid for most parts of the processed images. Lubk's paper proposed a method to deal with the vortex beam containing the phase singularity. The phase of the electron beam exiting the skyrmions does not have such singular portion - although the in-plane magnetic inductions show the singular point in bubble center the phase of exit wave is continue. Pure phase object assumption that was cited by Lubk is another literature (T. E. Gureyev and S.W. Wilkins, J. Opt. Soc. Am. A, 15, 579, 1998). Grureyev et al just used the phase contrast concept to introduce the image mode but not as the premise of deduction. In principle, the differential operation and the normalization by $I(z=z_0)$ in Eq. 1 remove the influence from the amplitude of the wave since the intensity of in-focus image presents the amplitude of the wave. Therefore Eq. 1 can give the proper phase solution.

$\nabla_{x,y}^{-2}$ is the inverse Laplacian operator, and it can be substituted by $-1/q^2$ in Fourier transform.

Allen's paper gave a deduction about this point. (Eq. 8-10 in "L. J. Allen, M. P. Oxley, Phase retrieval from series of images obtained by defocus variation. Opt. Comm.,199, 65-75, 2001")

The errors and inconsistence about q and k have been corrected.

3. English writing and wording.

- there are numerous typos and sometimes inadequate choice of words, e.g.

line 20: "magnetic moment" is misleading here, line 40 "need", line 48 "turns into" not appropriate in this context, line 64 "charity", line 116 "configuraiton", line 232, 238 "configure", line 263 "3D structure reconstruction" not the right term

The text needs very much polishing in general.

R: We carefully modified the manuscript and tried our best to correct the typo and grammar errors.

4. Fig.1b: Why there is a threshold in the simulated induction maps (black mask)?

R: The color maps in Fig. 1b is the TIE retrieved induction maps which are influenced by the regularization parameter q_0 . Thus a threshold, 5% of the maximum induction, is used to get rid of that effect.

5. Line 273: Please provide the value of q_0 in "1/nm" not pixels.

R: It was changed to $5 \times 10^{-3} \text{ nm}^{-1}$.

Reviewers' Comments:

Reviewer #1:

Remarks to the Author:

This is a much improved manuscript that now makes clear what has been done by the authors and by others in the past, and what is new. The major novelty of the paper is the three dimensional reconstruction of the magnetic induction. The authors have utilised the method of Wolf (reference 30) which first demonstrated how to accomplish the 3D magnetic texture of a Bloch skyrmion in FeGe using holographic vector tomography. That means that the authors can measure not just the vorticity of the skyrmion bubble but also its polarity and hence the chirality. This means that when a magnetic bubble switches its circular direction, they can say whether its chirality also changes. This is a degree of freedom which is not available to a skyrmion and its chirality is fixed by the Dzyaloshinskii-Moriya interaction and the orientation of the applied magnetic field.

The editor should decide whether this observed chirality switch in a skyrmion bubble in centrosymmetric MnNiGa is of sufficiently wide interest to a journal such as Nature Communications, or is more suited to a more specialist condensed matter or magnetism journal.

The few remaining comments about the revised manuscript are:

Fig 1a Why does the magnetization go to zero at a given radius? Are they magnetic disks?

What model was used to generate the skyrmions?

Fig 1b Why was the contrast below 5% of maximum magnitude masked in the magnetization mapping to filter the artifacts induced by the regularisation parameter in TIE?

Fig 4 There is a confusion of B and M between the figures and the caption.

Reviewer #2:

Remarks to the Author:

First of all, I appreciate that the authors have addressed all of my concerns and in general I am satisfied with their detailed reply and corresponding modifications of the main text. I do not doubt that the major findings namely the identification of the chirality and polarity of the type I bubble as well as the discrimination to the type II bubble is possible by inspecting the upper and lower surface of one component of the magnetic induction reconstructed by tomography. I also do not doubt the 3D reconstruction of the in-plane vector-field of one bubble (#2), one of the few which seemed to be stable for both tilt series.

However, by reading the revised text I identified remaining problems (in particular with respect to the papers' impact for the community):

1. The reason for the dynamic chirality flips (content of title!) is not resolved, because they were unintended. Maybe one reason could be a small in-plane field. This must be investigated further. For example, the authors mention in the last sentence of the abstract: "Our results offer valuable insights into the fundamental dynamics to understand the chirality behavior of skyrmion bubbles". But I miss exactly this understanding although it is extensive observation. Maybe also a micromagnetic simulation and corresponding energies could give more insight.

2. A way how to use the chirality (switching) information (e.g., as read-out for memory devices) and make it useful for technological applications is not discussed. As the authors mentioned the polarity (which could be easily read out) was not changed.

3. The central theme through the entire manuscript is not consistent. The title promises to study the reason for the chirality flips but it is not done. The abstract promises the 3D vector-field reconstruction of magnetic induction, i.e., all three components of all bubbles but in fact only two components of one bubble or one component of the other bubbles with two different states are presented.

4. Still the writing (wording) and presentation has serious flaws. In the following I mention only a few examples, but there are much more:

- the term Skyrmion bubble in general is questionable

- sentence line 20: "The induction configuration of the bubbles was determined from investigating

the magnetic vectors in entire space." contains strange combinations: "induction configuration" should be either "magnetic configuration" or "magnetic induction"

One suggestion: "The magnetic configuration of the bubbles was determined from the reconstructed magnetic induction (B-field) at their surfaces and their center."

- in Fig. 2 tilt axes should be indicated
 - line 134 Bx matrix wrong term better Bx array, By array
 - in Fig. 3 sub panel label must be in the upper corner
 - line 153 "serial tilting" wrong term, just say tilt series
 - line 157 "extension" must read: elongation (see for example Midgley and Weyland, Ultramicroscopy 96 (2003) 413, [https://doi.org/10.1016/S0304-3991\(03\)00105-0](https://doi.org/10.1016/S0304-3991(03)00105-0))
 - line 165 "out-plane" must be out-of plane
 - line 168 "thickness measurement error" minor comment regarding content: thickness measurement should not influence the tomographic reconstruction
 - in Fig. 4: What is shown in a and b upper left panel labeled with Mx, My
 - line 219: "That is to say, they cannot reveal the dynamic behavior of the bubbles in a field free condition." Bad English
 - line 221 "be against" should be "anti-parallel"
 - line 236 "always fabricated by focus ion beam (FIB)" not always, self-grown NWs are also possible
 - line 240: "is tilted around its short axis." One way out is to orient the needle-axis +- 45° to the tilt axis. (e.g. see Wolf et al., Commun. Phys. 2 (2019) 87 <https://doi.org/10.1038/s42005-019-0187-8>)
 - line 242 "prolongs" better "elongates"
 - lines 311,313, 314 "mappings" must read in these contexts "maps"
 - actually the whole sentence from lines 311-315 is very badly written.
- There are much more issues like these, also in the supplement.

In its current form I still cannot recommend the paper to be published in Nature Communications. In my opinion, it needs a second major revision.

Reviewer #1 (Remarks to the Author):

This is a much improved manuscript that now makes clear what has been done by the authors and by others in the past, and what is new. The major novelty of the paper is the three dimensional reconstruction of the magnetic induction. The authors have utilised the method of Wolf (reference 30) which first demonstrated how to accomplish the 3D magnetic texture of a Bloch skyrmion in FeGe using holographic vector tomography. That means that the authors can measure not just the vorticity of the skyrmion bubble but also its polarity and hence the chirality. This means that when a magnetic bubble switches its circular direction, they can say whether its chirality also changes. This is a degree of freedom which is not available to a skyrmion and its chirality is fixed by the Dzyaloshinskii-Moriya interaction and the orientation of the applied magnetic field.

The editor should decide whether this observed chirality switch in a skyrmion bubble in centrosymmetric MnNiGa is of sufficiently wide interest to a journal such as Nature Communications, or is more suited to a more specialist condensed matter or magnetism journal.

R: We sincerely thank the referee for carefully reading our manuscript and noting that the major novelty of our paper is the three-dimensional reconstruction of the magnetic induction. Following the referee's comments and suggestions, we carried out additional experiments and analyses. Moreover, we carefully and thoroughly revised the language in the manuscript. Below, we address the referee's comments and questions on a point-by-point basis. We hope the referee is satisfied with the revised manuscript and our response.

Fig 1a Why does the magnetization go to zero at a given radius? Are they magnetic disks?

R: Yes, we use a disk model to simulate the magnetic features of the bubbles. In general, DDI skyrmions can be stabilized by a sixfold symmetric array. To simplify the simulation, a single bubble was generated by the OOMMF code with a disk-shaped boundary.

Fig 1b Why was the contrast below 5% of maximum magnitude masked in the magnetization mapping to filter the artifacts induced by the regularisation parameter in

TIE?

R: In the TIE method, the regularization parameter q_0 is used to replace $\frac{-1}{|q|^2}$ with $\frac{-|q|^2}{(|q|^2+q_0^2)^2}$ to prevent the divergence of $\mathcal{F}^{-1}\left\{\frac{-\mathcal{F}[.]}{|q|^2}\right\}$; however, this parameter suppresses the low-frequency part of the original data. For example, if $q_0 = 2$, the strength of information for $q = 2$ may be 4 times smaller than the original value (1/4 vs. 1/16). However, for high-frequency information, such as $q = 10$, the influence is minimal (1/100 vs. 1/108.08). Thus, q_0 exaggerates high-frequency information. As described in our previous papers [Cui, J. et al. Artifacts in magnetic spirals retrieved by transport of intensity equation (TIE). *J. Magn. Mater.* 454, 304-313, (2018); Yao, Y et al. Magnetic hard nanobubble: A possible magnetization structure behind the bi-skyrmion. *Appl. Phys. Lett.* 114, 102404, (2019)], this distortion can lead to some artifacts in the results. Considering the noise in the experimental data, this distortion may also cause high-frequency noise. Therefore, a contrast threshold of 5% was employed to address this disturbance with a proper q_0 .

Fig 4 There is a confusion of B and M between the figures and the caption.

R: Thank you. This error has been corrected.

Reviewer #2 (Remarks to the Author):

First of all, I appreciate that the authors have addressed all of my concerns and in general I am satisfied with their detailed reply and corresponding modifications of the main text. I do not doubt that the major findings namely the identification of the chirality and polarity of the type I bubble as well as the discrimination to the type II bubble is possible by inspecting the upper and lower surface of one component of the magnetic induction reconstructed by tomography. I also do not doubt the 3D reconstruction of the in-plane vector-field of one bubble (#2), one of the few which seemed to be stable for both tilt series.

R: We thank the reviewer for his previous comments, which helped us to improve the presentation of the manuscript.

However, by reading the revised text I identified remaining problems (in particular with respect to the papers' impact for the community):

1. The reason for the dynamic chirality flips (content of title!) is not resolved, because they were unintended. Maybe one reason could be a small in-plane field. This must be investigated further. For example, the authors mention in the last sentence of the

abstract: “Our results offer valuable insights into the fundamental dynamics to understand the chirality behavior of skyrmion bubbles”. But I miss exactly this understanding although it is extensive observation. Maybe also a micromagnetic simulation and corresponding energies could give more insight.

R: Thank you for your helpful comments. As we mentioned in the manuscript, the field-free bubbles, which were obtained via the field cooling method, are in metastable states and can thus be manipulated by external stimulation. In practice, the electron beam can flip the bubbles during the initial observation; however, the bubbles were gradually stabilized after a shower period. We selected a region containing stable bubbles to acquire our data. During the data acquisition process, the investigated bubbles maintained their features, as shown in the SI. However, when the sample was removed from the LTEM for the 90-degree rotation for the next acquisition, the new beam illumination may change the configuration of some bubbles in the observation area. We assessed the influence of the electron beam in SI to elucidate this effect. We believe that the electron beam may induce a local temperature increase that may be responsible for this dynamic flipping; this result was also verified in the temperature control experiment. The sample was initially balanced during the first characterization; however, when the sample was removed from the TEM and rotated for the next observation, a new balance must be established during the second beam irradiation, as some bubbles may have different configurations. Unfortunately, because we lack additional evidence to explain the detailed process, we modified the title of the manuscript to “Chirality flips of skyrmion bubbles” to prevent confusion.

2. A way how to use the chirality (switching) information (e.g., as read-out for memory devices) and make it useful for technological applications is not discussed. As the authors mentioned the polarity (which could be easily read out) was not changed.

R: We thank the referee for raising this interesting question. To date, the key operation of skyrmion racetrack memory is to assist the DMI skyrmion in reading/writing with electrical currents. Inspired by this method, we propose a new mechanism that uses the spin transfer torque technique to select different chiral DDI bubbles. A constant current carries bubbles with uniform chirality and polarity passing through an electron beam, and the beam can only determine whether the chirality of individual bubbles is flipped and does not change the orientation of the polarity. Then, the bubbles enter different branches due to the Magnus effect. The sensors in each branch detect only the existence of the bubbles (whatever their polarity) when recording the information state. The electron beam can easily be shrunk to tens of nanometers to manipulate individual DDI bubbles rather than changing the entire spin configuration of DMI skyrmions.

3. The central theme through the entire manuscript is not consistent. The title promises to study the reason for the chirality flips but it is not done. The abstract promises the 3D vector-field reconstruction of magnetic induction, i.e., all three components of all bubbles but in fact only two components of one bubble or one component of the other bubbles with two different states are presented.

R: We apologize for the confusion. We understand the concerns of the reviewer. We evaluated chirality flipping by analyzing the 3D magnetic feature of the bubbles. After careful analyses, we recognized that it was not necessary to know all information about the bubble components to distinguish the type of bubble. Therefore, we focused on evaluating the 3D spin configurations of the bubbles after their chirality flipped. To clarify this point, the title has been changed to “Chirality flips of skyrmion bubbles”. In addition, we added new experimental data to provide a possible explanation for the flip; however, this topic should be studied further in future work.

4. Still the writing (wording) and presentation has serious flaws. In the following I mention only a few examples, but there are much more:

- the term Skyrmion bubble in general is questionable
- sentence line 20: “The induction configuration of the bubbles was determined from investigating the magnetic vectors in entire space.” contains strange combinations: “induction configuration” should be either “magnetic configuration” or “magnetic induction”

One suggestion: “The magnetic configuration of the bubbles was determined from the reconstructed magnetic induction (B-field) at their surfaces and their center.”

- in Fig. 2 tilt axes should be indicated
- line 134 Bx matrix wrong term better Bx array, By array
- in Fig. 3 sub panel label must be in the upper corner
- line 153 “serial tilting” wrong term, just say tilt series
- line 157 “extension” must read: elongation (see for example Midgley and Weyland, Ultramicroscopy 96 (2003) 413, [https://doi.org/10.1016/S0304-3991\(03\)00105-0](https://doi.org/10.1016/S0304-3991(03)00105-0))

- line 165 “out-plane” must be out-of plane
 - line 168 “thickness measurement error” minor comment regarding content: thickness measurement should not influence the tomographic reconstruction
 - in Fig. 4: What is shown in a and b upper left panel labeled with Mx, My
 - line 219: “That is to say, they cannot reveal the dynamic behavior of the bubbles in a field free condition.” Bad English
 - line 221 “be against” should be “anti-parallel”
 - line 236 “always fabricated by focus ion beam (FIB)” not always, self-grown NWs are also possible
 - line 240: “is tilted around its short axis.” One way out is to orient the needle-axis $\pm 45^\circ$ to the tilt axis. (e.g. see Wolf et al., Commun. Phys. 2 (2019) 87 <https://doi.org/10.1038/s42005-019-0187-8>)
 - line 242 “prolongs” better “elongates”
 - lines 311,313, 314 “mappings” must read in these contexts “maps”
 - actually the whole sentence from lines 311-315 is very badly written.
- There are much more issues like these, also in the supplement.

R: To address the language issues, the manuscript and the supporting information have been revised by one or more highly qualified native English-speaking editors at AJE. If the referee has additional sentences that they believe require revision, we would be happy to incorporate any suggestions.

In its current form I still cannot recommend the paper to be published in Nature Communications. In my opinion, it needs a second major revision.

Reviewers' Comments:

Reviewer #1:

Remarks to the Author:

I am now content with the paper to be published. The paper reports noteworthy results and hopefully will be a significant paper in the field. The authors have acted upon the comments from the referees and as a result the manuscript is much improved.

Reviewer #2:

Remarks to the Author:

The authors have severely revised and improved the manuscript further. I have only a few remarks left:

1.) I find the proposed concept of race track memory using DDI skyrmions interesting (new Fig. 5 and corresponding text), but I cannot judge the validity of it, because this topic exceeds my expertise.

2.) There are still some minor issues:

What are the units of the colorbar in Fig. 4? If the values are in A/m, then they are with $\sim 4 \cdot 10^6$ too high, because they correspond to ~ 5 Tesla!

Line 287: "spin transfer torch" must read "spin-transfer torque"

Line 358: I suggest to write: "Bx map was computed from the derivative of the phase in y direction."

Line 363: q is the coordinate or variable or spatial frequency in Fourier space, not the grid.

After especially my point 2) is addressed, I can recommend the paper for publication.

REVIEWER COMMENTS

Reviewer #1 (Remarks to the Author):

I am now content with the paper to be published. The paper reports noteworthy results and hopefully will be a significant paper in the field. The authors have acted upon the comments from the referees and as a result the manuscript is much improved.

R: Thank you for recommending our manuscript to be published in *Nature Communications*

Reviewer #2 (Remarks to the Author):

The authors have severely revised and improved the manuscript further. I have only a few remarks left:

1.) I find the proposed concept of race track memory using DDI skyrmions interesting (new Fig. 5 and corresponding text), but I cannot judge the validity of it, because this topic exceeds my expertise.

R: We sincerely thank the referee for careful reading of our manuscript, and for pointing out that race track memory using DDI skyrmions are intriguing. In fact, the race track memory using DMI skyrmions are extendedly studied by the theoretical and experimental group [Zhou Y, Ezawa M. A reversible conversion between a skyrmion and a domain-wall pair in a junction geometry. *Nature communications*, 2014, 5(1): 1-8; Zhang X, Ezawa M, Zhou Y. Magnetic skyrmion logic gates: conversion, duplication and merging of skyrmions. *Scientific reports*, 2015, 5(1): 1-8.], but it's rare for the DDI skyrmions. We hope our work might evoke more theoretical studies on the race track memory using DDI skyrmions and may also lead to the realization of skyrmion-based spintronic devices.

2.) There are still some minor issues:

What are the units of the colorbar in Fig. 4? If the values are in A/m, then they are with

$\sim 4 \times 10^6$ too high, because they correspond to ~ 5 Tesla!

R: The resolution of Fig. 4 in the PDF file is not good. The order of the value is $10^{(-6)}$, not 10^6 which represents the relative strength of magnetic induction deduced from the phase differential, but not the absolute value of the induction. However, in the revised manuscript, it has been changed to the absolute induction now. And the corresponding calculated magnetization is about 7×10^5 A/m, which agrees well with the measured value of 8×10^5 A/m [Bei Ding et al, Manipulating Spin Chirality of Magnetic Skyrmion Bubbles by In-Plane Reversed Magnetic Fields in $(\text{Mn}_{1-x}\text{Ni}_x)\text{65Ga}_{35}$ ($x=0.45$) Magnet, Physical Review Applied, vol. 12, 054060 (2019)].

Fig. 4 The magnetic spin transition of bubble #5. The B_x (a) and B_y (b) features in the top, middle and bottom x-y sections. The solid arrows indicate the direction of the corresponding component, while the dashed arrows indicate the speculative direction of the orthogonal component based on prior knowledge of Bloch skyrmion bubbles. The color legend indicates the magnitude of induction components in x or y direction, which corresponds the magnetization between $-6.5 \times 10^5 \sim 7 \times 10^5$ A/m and well agrees with the measured value of 8×10^5 A/m.⁴²

Line 287: "spin transfer torch" must read "spin-transfer torque"

Line 358: I suggest to write: "Bx map was computed from the derivative of the phase in y direction."

Line 363: q is the coordinate or variable or spatial frequency in Fourier space, not the grid.

After especially my point 2) is addressed, I can recommend the paper for publication.

R: We sincerely thank the referee for recommending our manuscript to be published in *Nature Communications*. In the revised version, we have modified above issues.